# Re-Evaluating Expertise: Principles for Food and Nutrition Security Research, Advocacy and Solutions in High-Income Countries

**DOI:** 10.3390/ijerph16040561

**Published:** 2019-02-15

**Authors:** Danielle Gallegos, Mariana M. Chilton

**Affiliations:** 1School of Exercise and Nutrition Sciences, Queensland University of Technology; Brisbane 4059, Australia; 2Center for Children’s Health Research, Institute of Health and Biomedical Innovation, Queensland University of Technology, Brisbane 4101, Australia; 3Department of Health Management and Policy, Dornsife School of Public Health, Drexel University, Philadelphia, PA 19104, USA; mmc33@drexel.edu

**Keywords:** food and nutrition security, research, values, co-creation, trauma-informed

## Abstract

Drawing on examples from Australia and the United States, we outline the benefits of sharing expertise to identify new approaches to food and nutrition security. While there are many challenges to sharing expertise such as discrimination, academic expectations, siloed thinking, and cultural differences, we identify principles and values that can help food insecurity researchers to improve solutions. These principles are critical consciousness, undoing white privilege, adopting a rights framework, and engaging in co-creation processes. These changes demand a commitment to the following values: acceptance of multiple knowledges, caring relationships, humility, empathy, reciprocity, trust, transparency, accountability, and courage.

## 1. Introduction

Food insecurity is a symptom of our social, economic, political, and ecological systems in crisis. Hunger is not due to a lack of food production or availability but rather to the unequal and unjust distribution of people’s entitlements to social and economic support [1]. The economically, politically, and socially powerful also control access to food and conditions under which food is available, effectively limiting the capabilities of others. These crises have at their root the continued legacy of colonization and the overarching neoliberal principles of the market economy and personal responsibility. These conditions perpetuate the structural and social institutions that undermine individual and collective agency. The result for people who are low-income is limited access to healthy food and other basic needs such as safe and affordable housing, utilities, gainful employment, and opportunities for political and civic participation. 

Rising obesity rates across all social strata, overall low breastfeeding rates, and continued disparities in food insecurity point to systems failures and to inadequate approaches to improve nutrition and food security. Included in these failures is the lack of engagement with appropriate experts with lived experience. Experts from dominant classes have become adept at aligning with powerful authorities in order to interpret and translate complex issues into “health-speak” while viewing people who lack income as passive recipients of expert nutrition and financial knowledge. In high-income countries a primary response to food and nutrition insecurity has been the growth of the charitable food sector, while government and public support for adequate wages and entitlements to basic needs, adequate means for earning money, and a publicly funded safety net have been receding or are under threat [2,3,4]. The lack of success in addressing food and nutrition insecurity indicates that there is a serious gap between supposed knowledge sitting with the “experts” from academia, law, non-governmental organizations (NGOs), corporations, and other arenas of social and political power, and the realities of people who struggle with food insecurity. This gap is an indication that experts with financial resources and power do not truly understand the causes and experiences of food insecurity, and thereby promote solutions that are misplaced or inadequate. 

While there are many examples of people with lived experiences with poverty and food insecurity that are active in academia in a way that informs their work and strengthens their approaches, the academic research community overall has failed to effectively work with and learn from people who have lived experience in a manner that can promote lasting change [5,6]. Though there are several inspiring exceptions such as The Food Action Research Centre (FoodArc) in Nova Scotia, Canada and the Poverty and the Social Exclusion Program in the United Kingdom, the tendency in the academic study of food insecurity is to drown out, exclude, or marginalize the experiences of people with lived experience [7,8,9]. Additionally, those with lived experience with food insecurity within academia can help to lead the way for researchers and others, yet due to potential stigma and structural barriers, they may not be willing to do so [10]. As there are growing numbers of people who have experienced poverty and also report food insecurity during college years [11,12], engaging with people with lived experiencing in food insecurity in all arenas will strengthen and inform solutions that have otherwise been lacking. 

We encourage researchers, policy makers and non-profit organizations to ensure that the lived experience and wisdom of those who experience food and nutrition insecurity, including those in academia and other professional occupations, are central in the conceptualization of food and nutrition security challenges and solutions. We identify some challenges for doing so. Focusing our efforts on governments, social services agencies, NGO’s, and civil society (rather than on public private partnerships that engage the corporate sector) we characterize ways in which experts of all kinds can work together to identify the local, regional and national solutions that lead to effective nutrition and food security. 

This paper emerges from research undertaken by the authors working in different paradigms (nutrition and anthropology) with individuals and communities that are economically oppressed which include but are not limited to; indigenous peoples, migrants and refugees, and those experiencing hunger, poverty, and trauma. We acknowledge that we are both white and privileged; and that we are products of and operate within the colonialist structures of education, health and welfare. Our context likely limits our viewpoints and clouds our own understanding of what we have learned so far about solutions to food insecurity. We outline here what we hope can be the beginning of a dialogue about our own limitations and the limitations of the research community. We start with our experiences in addressing poverty and food insecurity through our lenses as people who have had the privilege to work with families and communities that have experienced food and nutrition insecurity. Gallegos has worked as a public health nutritionist among Torres Strait and Pacific Islander communities, migrant and refugee communities, and marginalized youth in Australia. Chilton has worked with the Southern Cheyenne and Arapaho tribes in the United States and with caregivers of young children participating in public assistance in the United States who are primarily African American and Latinx. 

First, we identify the significance of working in partnership with experts with lived experience of food insecurity, we then address the challenges to collaboration and co-creation, and finally we describe the necessary principles and values that can help to drive potential solutions.

## 2. Examples and Insights on Benefits of Shared Expertise 

We are aware that there are many types of programs that have partnered with people who know food insecurity and hunger first-hand with indigenous groups, farmers and community activists [13,14]. However, in the interest of utilizing our own experiences as grounding for our conviction that partnership is key, we focus on specific examples from Australia and the United States. The Australian example provides insight into how co-creation of solutions can be developed in programs already prescribed by health and political structures. It could be argued that the solutions developed in this program were expedient and immediate, framed by the structures in which the program was embedded. The U.S. example demonstrates the additional step around developing capacity for political action that go to root causes of food insecurity such as violence and discrimination, and the systems that perpetuate these dynamics.

### 2.1. Australia

Good Food for New Arrivals (GFNA) was a nutrition intervention program funded by the Commonwealth Department of Health through the national child nutrition program and the Department of Family and Community Services from 2001 to 2008 [15,16]. The original aim of GFNA was not to address food and nutrition security but rather to develop nutrition resources to “educate” newly arrived refugee families about nutrition within the western context. Originally the program set out to change what were unhealthy food choices as determined by nutrition and health promotion professionals. Rather than rely on this second-hand knowledge, the program undertook a community participatory approach that engaged members of identified communities (South Sudanese, Hazara Afghani, African ‘Grand Lacs’ (Democratic Republic of Congo, Rwanda, Burundi), Iraqi and Iranian [17]. Community members identified iron deficiency, poor appetite in children, food safety and foods appropriate for school as key issues. GFNA was also the first program to identify that food and nutrition insecurity was an issue for refugees settling in Australia with 70% of households running out of money for food [18]. Over the program’s duration GFNA developed a set of resources that addressed multiple issues identified by both communities and health professionals. However, an evaluability assessment of the program identified that the underlying funding premise was that refugees were “doing something wrong”. After engagement with communities GFNA identified that the deficits lay within the infrastructural constraints of the system and with health professionals [19]. This realization led to the identification of a broader range of activities including the development of nutrition champions from within communities and influencing system changes such as the speed at which welfare payments were processed on arrival. The examples of the reasons for running out of food clearly demonstrated a link to trauma and adverse childhood events and included: high medical costs associated with amputation due to a landmine, having family back in the country of origin and feeling guilty about eating, and also moving from having no food to having some food [18].

### 2.2. United States

Witnesses to Hunger (Witnesses) is an ongoing participatory action program that works with women who know hunger first-hand to increase their meaningful participation in the national dialogue on poverty. Witnesses began in Philadelphia in 2008 with 42 mothers of young children that then expanded to multiple cities to reach over 100 participants. Most members of Witnesses were eager to share their experiences of poverty, their ideas on ways to overcome it, and to inform key decision-makers about the importance of improving labor laws, neighborhood zoning codes, education, tax and labor policies, and to recognize the true value of each person, of motherhood, childhood, and family struggle. Utilizing a human rights approach where the rights-holders participate in shaping the problem, challenges, and solutions, members of Witnesses to Hunger have not only contributed to ethnographic and qualitative research, but also mounted over 30 exhibits of their photographs in locations such as the US Senate, the US House of Representatives, city halls and state houses for audiences that include elected officials, federal, city and state agency administrators, community leaders, the press and the lay public [20,21]. Exhibits also include public forums, hearings and formal testimonies with elected officials. In addition to exhibits, policy briefs, individual and group visits to elected officials, members of Witnesses launched their own blog series and developed a social media presence. They speak at conferences and have co-authored scholarly publications and newspaper opinion essays that demand focus on root causes of food insecurity such as violence (institutional racism, community violence, interpersonal violence, and policy violence), discrimination, and inadequate health and welfare systems. 

### 2.3. Benefits and Insights

From these two case studies of co-creation and mutual engagement, there emerged four significant insights: (1) solutions should recognize personal and collective agency and seek to promote freedom and opportunity; (2) complex issues are dependent on policy change across interlocking systems, and solutions therefore need to be broadly conceptualized across and not within systems (e.g., political, health, economic, welfare); (3) root causes as identified by experts with lived experience should inform the solutions; (4) and co-creation efforts require building trust and transparency. 

First, the immediate solutions usually generated by researchers and advocates alike are those that are “top-down” that view the food insecure person, family or community as passive recipients of assistance. For example, many researchers suggest that if we improve public nutrition assistance programming, or seek to improve other aspects of the safety net such as improving access to housing vouchers or healthcare, people’s lives will improve [22,23,24]. This was the case for GFNA, although there was an intent to build capacity in developing “nutrition champions”, the onus was on improving access to the elements of the current “broken” system. This system was filled with delays in getting access to income and food resulting in an increased reliance on individuals and organizations to fill the gap. The engagement with Witnesses identified that policy solutions needed to go beyond simply “improving the safety net.” Members of Witnesses viewed the safety net as an untrustworthy system that remains broken, inadequate, and undesirable. Members of Witnesses did not want to receive more government assistance; rather, they had a strong desire for freedom and opportunity in developing more entrepreneurship, improving access to education for themselves and their children, and to safe neighborhoods which included ridding their neighborhoods of drug dealers and users, and greater investment in public services such as improved playgrounds, blight alleviation, garbage pickup, and other opportunities for neighborhood improvements. 

Secondly, GFNA and Witnesses identified that food and nutrition security was not just the remit of a single system but involved policy change across systems. Food and nutrition security are not simply about lack of food but are an indication of a failure of income, housing and health systems to deliver. Members of Witnesses were eager to learn about how to shape policy. Yet, as training was provided to those who were interested in advocating for solutions, the members quickly discovered that available policy solutions were too siloed. They preferred approaches to be more holistic. For instance, they saw a direct relationship between the trade-offs of paying for food and housing, and therefore, they wanted to advocate for programs that incentivized higher paying jobs and entrepreneurship, so people could pay for their own food and market rate rents. Their frustration with the official policy process was tangible, and they have mostly abandoned standard policy-related solutions and turned their attention to more home-grown solutions that involve neighborhood clean-ups, clothing exchanges, and improved access to local housing. 

Thirdly, members of Witnesses have insisted that food insecurity was *not* their most significant issue; whereas, exposure to violence and lack of safety were the central problems that tied all other problems together [21,25]. This was also evident for GFNA. But despite the program being sponsored by a torture and trauma agency, food and nutrition was effectively compartmentalized away from these issues. For Witnesses however, the insistence on the importance of safety led the research team into a new area of research and policy focus on exposure to violence and trauma [26,27]. With this new knowledge about the centrality of trauma and adversity, and the need for individual and collective resilience and, holistic, group-oriented approaches to social services, the research team developed a new intervention effort called the Building Wealth and Health Network (The Network). The Network works with caregivers of young children through a trauma-informed peer support approach (to address exposure to violence), financial empowerment education and new savings accounts, where people’s savings are doubled (to address economic insecurity) [28,29]. The Network has reduced the odds of economic insecurity and improved mental health and income. Without that intentional and long-term engagement and magnanimous expertise of members of Witnesses, they would never have been able to develop effective solutions.

Finally, while GFNA undertook a participatory approach and there was recognition of individual and collective agency, the capabilities of community members were not fully realized. On reflection, part of this was a failure of those in power to fully trust experts with lived experience and their conceptualization of the issues and the solutions. This lack of trust often masqueraded as lack of time to develop partnerships, difficulty in engaging individuals and communities, as well as empathy regarding the overwhelming number of issues community members faced. The project officers undertook a wide range of activities and advocacy on behalf of the community members [15,16]. On the flip-side, GFNA was one of the first projects to employ members of the community as project consultants in order to provide cultural and experiential expertise (previously community members were expected to volunteer their services). In Witnesses to Hunger, the long-standing nature of the relationship between members of Witnesses (primarily Black and Latinx women) and the research team (racially diverse, with majority white leadership in terms of funding and decision-making) engendered some feelings of mutual trust and accountability, especially as all engagement by members of Witnesses was treated as professional work for which members were paid market-rate wages and honoraria. However, partnering across racial barriers and the spoken memory of generations of mistrust, misunderstanding, and oppression among black women by white women has generated ongoing challenges that bring to light questions about racism, leadership, and misaligned priorities, mission and goals. 

Throughout both of these examples of partnered research it is clear that the best solutions are not simply based in science and standard empirical evidence, but also in what Maria Miess asserts is the wisdom that comes from experience and struggle [30]. 

## 3. Challenges of Sharing Expertise and Co-Creation of Solutions

There are many challenges to sharing expertise among traditional, highly educated and well-resourced experts (this includes those with lived experience that adopt a traditional scientific approach) and experts who have lived experience and who do not share the tools, resources and power of academia. These include the refusal to look at food insecurity as related to social factors such as (1) historical and contemporary racism and discrimination, (2) the culture of academia, (3) siloed thinking, and (4) marginalization and cultural differences in meaning.

### 3.1. Historical and Contemporary Racism and Discrimination

Our first challenge to overcome is our lack of willingness to identify the discriminatory social structures that cause poverty, deprivation, and trauma. As a single example among so many for African Americans, in 1898 with the publication of the Philadelphia Negro, sociologist William Edward Burghardt Du Bois identified how the struggles within the black community—poor health, unemployment and deplorable living conditions—are due to racial segregation. These directly stem from a socially constructed racial hierarchy that isolates, segregates and disenfranchises black people. He asserted that segregation and discrimination results in devastating poverty. The 1968 Kerner Commission Report described that the single most important issue for the struggles of African American people in terms of housing, poor nutrition, poor health, and low educational attainment in the United States is that whites systematically discriminate against and marginalize people of color [31], for example in its municipal, city, state and regional housing policies, in media coverage, and in general American society. Yet nutrition and food security researchers in the United States continuously ignore the dynamics of racism and discrimination that underlie poverty. Against this backdrop, it is only in the past few years that researchers in nutrition are beginning to call out discrimination and lack of equity as significant to the experience of food insecurity, obesity, and other nutrition-related conditions. The multiple forms of discrimination and oppression (systemic, interpersonal, structural, historical, etc.) are difficult to measure, and often do not fit in a simple model, yet researchers have begun to identify how lifetime, historical and systemic exposure to racism as associated with food and nutrition insecurity [32,33,34]. 

Australia has also demonstrated repeated failure to significantly address the blatant harms caused by discrimination. The same year as the Kerner Commission, William Edward Hanley Stanner delivered the nationally acclaimed Boyer Lecture “The Great Australian Silence” which argued that the history of invasion and the theft of lands and the genocide of Aboriginal and Torres Strait Islanders has been ignored [35]. Since this time, the Australian state has generally continued to ignore harms committed against Indigenous peoples [36]. In 2008 the Closing the Gap initiative was launched following a Social Justice report identifying serious inequity in health and life expectancy between Indigenous and white people. For instance, there are much higher mortality rates among Indigenous infants, and Indigenous men are dying more than 10 years earlier than their white counterparts [37]. Clearly, Australian health professionals failed to address the underlying structural power imbalances, intergenerational trauma and racism contributing to poor health [38,39]. Over time, nutrition researchers have all highlighted the inequalities related to food and nutrition for Aboriginal and Torres Strait Islander peoples and have identified the role of colonialism on the quantity and quality of food for Indigenous communities [40,41,42,43]. However, most are still describing the problem rather than identifying the root causes, that is, institutionalized racism, discrimination, poverty and marginalization. 

### 3.2. Culture of Academia

The culture of academia and the legacy of western European influences in scientific investigation creates blind spots that allow for scientists to ignore or obfuscate how discrimination shapes economic insecurity, illness and health. The definition of food insecurity itself—the lack of access to enough food for an active and healthy life due to economic circumstances [44]—lacks connection to social and political circumstances such as lack of access to living wages, lack of political power of people who are low-income, and to discrimination and exclusion. An improved definition would draw attention to context behind “economic circumstances” to include concepts of “economic exploitation” and “marginalization” that demonstrate how food insecurity is a concept that is in relationship to societal dynamics. Despite Krieger’s 1994 call to action for epidemiologists to move beyond biomedical individualism to acknowledge how health and disease have their roots in history, social relationships and political structures [45], food insecurity research published in English language research journals has been mired in the risk-exposure binary that still dominates health research. As Zuberi and Bonilla-Silva assert, researchers continue to ignore the large societal conditions that drive poor health and poor nutrition, and ignore their own place in perpetuating those conditions [46]. 

Most research and funding for research emanates from universities, well-resourced public policy centers, and from government sources, where a majority of people who are carrying out the research, making funding decisions, and generating research questions and methods are people without lived experience, and who do not see how the systems in which they are involved (i.e., education and government) are perpetuating poverty. While it is common knowledge in research circles that among food insecurity and poverty researchers there are people who have lived experiences with poverty, it is unclear how many there are, as this has not been previously studied or counted, and it is possible that such researchers would not readily describe these experiences due to real and perceived stigma as mentioned above. The culture of academia could also improve to be more accepting and inclusive of research scientists with lived experiences to help deepen understanding of the experiences and emphasize the importance of innovative approaches grounded in experience. Class, race and gender inequities are not solely among individual researchers, but are also built into institutional practices. As an example, universities and health systems have a long history of causing gentrification that further isolates people of color and people who are poor from mainstream resources [47]. Additionally, scientific methods demand strict definition of measurable problems, and center around testing of hypotheses of limited measures of covariates and outcomes. The pressures of producing peer reviewed research to establish academic credibility, and incentives for promotion in academia that utilize metrics unrelated to impact or how much engagement and authentic collaboration there is with research participants both work to devalue, isolate or discourage participatory research. Additionally, there is little to no incentive or recognition of the intensive time and trust building processes necessary for effective participatory research [48]. Overall the glorification of mainstream science and the pressures of academia to publish scientific research prioritize only one way of knowing about social problems such as food insecurity. This has led to a lack of appreciation for lived experience and wisdom from the streets, the farm, the reservation, and the neighborhood.

Qualitative researchers may view themselves as less engaged in research that ignores broader contexts of discrimination and inequality. However, we argue that most qualitative research still relies on a one-way process that extracts stories and experience from people who have lived experience, that generally enhances the investigator’s career through publishing books and articles, while those studied remain unseen, without political power, economic security, and legal recourse. We suggest that research move beyond the relatively simple process of gathering insight and stories from individuals and groups, and move into the co-creation of understanding and move to mutual problem solving in partnership. There are strong traditions from which to draw, such as action research by Sol Tax, applied anthropology, critical participatory action research [49,50,51], and indigenous methodologies where knowing and knowledge is built through relationality via yarning circles (Australian) [52], *talanoa* (Pacific/Maori) [53]. These methodologies are conversational techniques that involve the sharing of stories. They have at their core equal respectful engagement and the co-creation of knowledge [54]. 

### 3.3. Siloed Thinking

Reducing health inequalities and addressing social determinants of health requires greater integration across government and civil society. Sir Michael Marmot and colleagues have identified that action however is limited by organizational boundaries and “siloes” [55] (p. 86). Additionally, in most neoliberal high-income countries, funding streams are aligned with discrete government agencies that are based on outdated systems. In the US, it is well known that funding streams cannot be easily merged or braided together without acts of Congress. Even when federal agencies seek to work together, there becomes a territoriality of concern regarding programs where agency leaders are afraid of losing funding if they share some of their funding with other programs [22,56]. In Australia and the UK there is strong rhetoric about “whole-of-government” approaches for persistent social and environmental challenges. However, the three primary barriers for horizontal governance or “joined-up” government are identified as: a deeply entrenched program focus based on funding streams that remain siloed, centralized decision-making that undermine devolved decision-making, and the reliance on co-locating services rather than adjusting the underlying operating systems [57,58]. Overall, this siloed thinking is what Rebecca Costa refers to as a “super meme”—a way of thinking that is simply accepted despite the fact that it is irrational—that stymies innovative action to solve society’s most intractable problems such as hunger [59]. 

### 3.4. Marginalization and Cultural Differences in Meaning

The poor are marginalized or excluded by multiple systems such as zoning laws, school funding laws (as in the US), higher education discrimination, housing discrimination, as well as the systems of academic inquiry. An example of this is how from the perspective of academic researchers, sex workers, people who are homeless, or people who are “disconnected” from public welfare programs are considered “hard to reach” because of recruitment methods that have limited timeframes during the day, extensive costs to the individuals, and inadequate community engagement [60]. Additionally, food insecurity researchers tend to look at food insecurity over a short period of time, usually in cross-sectional studies, or in a one-two year time frame, with a few exceptions. Yet some groups have a very different view. Indigenous peoples may view their experiences with hunger and ill-health as stemming from times of genocide of their peoples and through ongoing injustices of broken treaty rights (where treaties exist) or failure to recognize sovereignty through constitutional reform [61,62]. Additionally, they may consider hunger to be an issue consistent with the violation of their sovereignty and the rights of nature [63]. 

## 4. New Principles for Food and Nutrition Security Research

While the challenges mentioned above are serious, they can be overcome through new ways of thinking about our work, and through adopting core principles and values. Developing solutions for food and nutrition insecurity, particularly when the issue involves marginalized groups in high-income countries. New approaches require a change in mindset and in our ways of working. We argue here for a set of core principles and values that should underpin and inform our actions in research and in devising evidence-based solutions. These principles draw from a variety of traditions such as civic agriculture and civic dietetics, trauma-theory, emancipatory education, and indigenous worldviews. Learning from these frameworks we propose four core principles that should guide our work in alleviating household food insecurity: (1) a critical consciousness that requires individuals to constantly question their own and others’ positions; (2) working to deconstruct white supremacy; (3) a rights-based approach that ensures engagement with people with lived experience, and (4) actively engaging in co-creation processes where power is shared and all expertise is regarded as meaningful. 

### 4.1. Use Critical Consciousness and Emancipatory Processes That Transgress Boundaries

Critical consciousness lies at the heart of working “inside out” to question perceptions of ourselves, of privilege and of the social and structural institutions that seek to maintain divisions in society. Critical consciousness highlights the need for a reflexive approach (that is not just thinking but also acting) to understand and change inequities in power and privilege. It requires reorientation to a commitment to love for humanity and social justice [64,65]. This raising of consciousness leads to what Freire described as engaged discourse, collaborative problem-solving and a re-humanization of our social relationships. Critical consciousness is required to understand that marginalization is not inherent within an individual but is rather a result of the structural and social forces that create that lived experience [66]. For example, just focusing on the food insecure individual or household locates the marginalized person as “the problem,” whereas the problem is located within the social and political context. 

While Freire’s original conceptualization was focused on those who were experiencing oppression and or marginalization, increasingly it is being applied to all types of participants in social programs including the researcher or those involved in developing and delivering programs [67]. Critical consciousness therefore has three essential elements: critical reflection—an analysis and rejection of the social inequities that limit agency and contribute to poor health and wellbeing; political efficacy—the perceived capacity to effect social and political change individually or collectively; and critical action—the actions taken to change aspects of society that are unjust [68]. Integrating feminist and intersectional approaches is also important to integrate attention to multiple, intersecting identities such as race, gender and sexuality, that consider a whole person approach, and that puts the authority of lived experiences at the center of inquiry. This approach helps all experts involved to transgress boundaries of race, gender, age, class, sexuality and beyond to resist and subvert patriarchal oppression and white supremacy [69]. These practices can be put into action in ongoing nutrition education efforts, participatory action endeavors, and other types of qualitative research efforts. In Witnesses to Hunger, these principles were utilized in every group meeting, where members were invited to explore from their own experiences how policies fell short, and worked together to identify their ideas for solutions that were then crafted into the exhibits, information booklets, and postcards for bringing along to meetings with legislators. This community self-empowerment education approach is also utilized by the Poverty Truth Commissions that were first established in Scotland and spread to other cities in the UK, currently hosted and promoted by Church Action on Poverty [70,71].

### 4.2. Utilize an Anti-Oppression Framework and a Trauma-Informed Lens

An anti-oppression framework is one that seeks to undo the effects of oppression, oppose the roots of all forms of oppression, and to adopt an emancipatory approach to social change. Anti-oppressive practice has penetrated a variety of fields, and became most highly developed in social work practice and psychology, where attention to breaking down status quo definitions of identity, eliminating boundaries of social division based on gender, race, ethnicity, age, and other identities can help to improve the therapeutic relationship as well as bring about societal change. The approach also seeks to call attention to power differentials in our relationships. An anti-oppressive approach demands not only the practice of actively seeking to ensure we do not oppress others, but also to continuously recognize our own roles in perpetuating our privilege and power that can lead to oppression. This means actively taking a decolonization stance, employing approaches and methodologies that disrupt and reverse the ongoing exploitation and subjugation of people who have been marginalized, excluded and oppressed [9,53]. 

The intergenerational and interpersonal trauma associated with colonialism and imperialism and the vast arrays of “isms” and phobias such as racism, sexism, ableism, classism, homophobia, and xenophobia, can be acknowledged and addressed in our everyday actions and in our policy proposals. Trauma informed practice realizes the widespread impact of trauma, and pathways to recovery, recognizes symptoms of trauma in clients, participants, families, staff, systems, and in ourselves, responds to fully integrate trauma-knowledge to improve and inform policies procedures and practices, and actively resists re-traumatization [72]. Given decades of evidence that exposure to violence such as intimate partner violence, child abuse, neglect and other adverse childhood experiences, suicide attempts and ideation, and post-traumatic stress disorder are strongly associated with household food insecurity [26], taking a trauma-informed lens to co-creating solutions is fundamentally important. 

A trauma-informed approach to self-organization was essential in the methods of Witnesses to Hunger [21], and later created the foundation of The Building Wealth and Health Network [73], which significantly reduced food insecurity. Other trauma-informed approaches are being integrated throughout many school districts across the United States [74,75], and there are more calls for trauma-informed policy-making [76,77]. 

### 4.3. Utilize a Human Rights Approach

The right to food and to be free from hunger are fundamental human rights in the Universal Declaration of Human Rights and in the International Covenant on Economic, Social and Cultural Rights (ICESCR). It incorporates being able to have access to culturally appropriate and healthy food in order to live a healthy and fulfilling life without fear. Viewing food security as a human rights issue means that good nutrition should not be left to benevolence or charity, relegated to the remit of the charitable food sector. Instead food security should be respected, protected, and fulfilled by governments and NGOs to promote the health, security, and wellbeing of all people [78,79]. In order to advance the right to food, it is necessary to ensure that there is a national plan to respect, protect, and fulfil the right to food, and a comprehensive approach to ensure participation of many stakeholders (especially those who are most affected by food insecurity), in the development of solutions, as well as for redress and repair when the right to food is violated. Having a national measurement mechanism for monitoring and accountability is essential. The governments of Australia, UK and New Zealand (among others) fail to regularly monitor food insecurity and issues related to access and provision of food. This keeps comprehensive solutions to food and nutrition insecurity unknown or ad hoc. While national monitoring is not the only way to get started with a human rights approach, it can help to provide information and empirical evidence for monitoring and evaluation of interventions. 

Adopting a rights framework is a key tenet of the recommended approach to ensure:-Solutions ensure equitable access to nutritious food regardless of one’s circumstances;-Solutions move beyond charitable approaches to those that address capabilities and enhance individual freedoms to achieve health and wellbeing;-Trauma and stigma are not inflicted or exacerbated and healing opportunities that build resilience are integrated into food-related programming;-Food sovereignty is respected and promoted;-Policy development does not exacerbate inequalities or contravene other human rights in recognition that all human rights are universal, inalienable, indivisible, interdependent and interconnected;-Rights holders have a central role in bringing about solutions.

Advancing human rights, and the right to food is very challenging. This is especially true in high income countries such as Australia and the United States, where there are major cultural assumptions by a powerful elite that trivialize and downplay the importance of economic and social rights [3,78,80]. Additionally, the focus on food, rather than on the social, economic and political conditions that cause food insecurity limit the understanding and adoption of the rights framework. The emphasis on charitable food provision, the slow dismantling of an already inadequate social security safety net, and reliance on “trickle-down” economics to alleviate poverty are serious obstacles to helping civil society adopt a rights framework and to demand right to food [78,81,82]. Despite these challenges, if the research and advocacy communities in high income countries could begin to adopt a broader justice framework, and promote such solutions among advocates, the press, and policy-makers, it can help support the current efforts of civil society to ensure people can be empowered to demand their right to food, and to health and wellbeing. Efforts such as the Participation and the Practice of Rights (PPR) in Northern Ireland, Detroit Black Community Food Security Network and the Southern Rural Black Women’s Initiative for Economic and Social Justice (SRBWI) and other organizations of the US Human Rights Network, the international campaign La Via Campesina which advocates for food sovereignty, and the Right to Food Coalition in Australia, are just a few of many examples where economic social and cultural rights are being advanced by civil society despite the above-listed challenges. The research community has much to learn from these ongoing efforts.

### 4.4. Seek Co-Creation of Problems and Solutions

There is growing recognition in research circles that there needs to be a different paradigm of knowledge production [83] and a fundamental shift from privileging experimental expertise to experiential expertise [84]. Characterizing problems from the perspective of the scientific “expert” is using knowledge as a form of “discursive power in ways that privilege some definitions of health and social problems and marginalize others” [85]. If change is to occur, those in positions of knowledge “expert” status need to reorient their inquiries from describing the problem to research that seeks to understand the effectiveness of interventions [85]. 

In undertaking collective processes of inquiry, empowerment and action, the experimental and experiential experts need to work to remove power differentials and utilize their respective strengths to co-create a mutual understanding of: the life-world, the dispositions and aspirations of those who live in that life-world, the problem as socially-constructed and the solutions that will be best fit in that context [86]. In taking this approach we agree with other scholars that it is no longer possible or ethical to separate the “research” from the ensuing policy discussion. In the case of food and nutrition security the understanding of the problem and the solutions requires both experimental and experiential experts to lend their voices to ongoing policy discussions [49,85]. Indeed as Fine indicates, “it is the obligation of the scholar to not only expose social injustice but to transform unjust conditions” [49] (p. 116). Examples of such co-creation are efforts by Witnesses to Hunger, and the Poverty and the Social Exclusion in the United Kingdom research project funded by the Economic and Social Research Council consisting of collaboration between the University of Bristol (lead), Heriot-Watt University, The Open University, Queen’s University Belfast, University of Glasgow and the University of York [8,87]. 

## 5. Values for Sharing Expertise and Co-Creation

Underpinning the principles are a set of underlying values informing approaches to researching and programming for food security. Table 1 characterizes the values that inform this process and help to create an ethos of action that questions the status quo, empowers partnership development and makes use of different forms of expertise. These qualities require self and political awareness. 

These values transcend any one field of study and action and can provide some grounding to to continue to address the challenges of racism and discrimination, the limited culture of academia, siloed thinking, and marginalization and cultural differences in meaning and time horizons.

## 6. Conclusions

Co-creation of solutions and sharing expertise across boundaries of race, class, education level, gender and age are beneficial and necessary for devising meaningful, effective and lasting changes in food and nutrition insecurity. The co-creation of solutions on food and nutrition insecurity will however not come easily. The challenges of racism and discrimination, the culture of academia, our siloed thinking, and cultural differences will consistently be in the backdrop of our efforts, and may actively get in the way of creating and then implementing solutions. We propose here some organizing principles and values to help overcome these challenges. Without actively engaging with these, researchers may be perpetuating inequality and injustices that drive poor nutrition and health. Embracing multiple forms of knowledge, humility and courage, among many other values, may be difficult and unrewarded currently in our own spheres. Yet we suggest that the rewards of improving food and nutrition insecurity for millions far outweighs the discomfort many of us might have with shaking up and altering our ways of doing. We invite the rest of the food insecurity research community, especially those with lived experience, to weigh in on these principles and values, and we hope they will join us in establishing a shared international consensus for co-creating solutions that promote the right to food, and promote health and wellbeing for all.

## Figures and Tables

**Table 1 ijerph-16-00561-t001:** Underlying Values for Sharing Expertise.

Value	Description
Knowledges	Recognize that knowledge comes in a variety of forms and is not limited to book learning and the scientific method. Different forms of knowledge extend to different forms of expertise. Each participant brings a unique set of expertise to problem identification and solution creation that can be brought together to construct new knowledge.
Relationships	Build relationships that are genuine and long-lasting. These relationships need to be built on trust, reciprocity and an understanding of and explicit attention to differences that create power inequities.
Humility	For those with the power, education and privilege it is essential that we express an understanding of how our unearned privilege and societal rank limits our skill sets, and that these skills are not necessarily better than those of others. Coming to the work with humility and a beginner’s mind helps to undo power differentials based on education, gender, sexual orientation, economic resources, race, class, cultural background and spiritual beliefs.
Empathy	Build a powerful imagination in order to understand the life situations of others in order to be able to respond to social inequities. Empathy requires an understanding of the differences between self and other and an ability to understand and relate to another’s perspective, emotion and experience.
Reciprocity	Exchange material resources, ideas, social obligations and power for mutual benefit. Reciprocity is fundamentally steeped in conceptualizing balance and an interconnectedness across time and space. Reciprocity requires giving and receiving.
Trust	Trust is premised on respect, transparency, accountability and reciprocity. There needs to be mutual trust in the process and outcomes of the co-creation of knowledge and solutions.
Transparency and accountability	Recognize that there are mutual accountabilities for individuals and organizations. There may be accountabilities to education institutions, funders and donors, political ideologies, families, communities, and cultural traditions. There will be tensions between these accountabilities but in order for trust to develop transactions and encounters need to be transparent. In this way the primary accountability is to social change and to the disruption of institutional and social structures that maintain inequity.
Courage	Understand that to work in a different way, to be politically active and to challenge the status quo takes self-knowledge, fearlessness and a willingness to be vulnerable and uncomfortable.

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
