# Peer review of "Re-Evaluating Expertise: Principles for Food and Nutrition Security Research, Advocacy and Solutions in High-Income Countries"

_ijerph, 2019, doi:10.3390/ijerph16040561_

Round 1
Reviewer 1 Report
The paper represents a novel, thoughtful, and well-argued contribution to the literature on the topic of food security. While some elements of the key arguments are well-rehearsed ,such as the imperative to ensure the lived experience of those who are food-poor are central to inquiry and research into the topic, the linking of the insights from the US and Australian "case studies" with proposed values for the sharing of expertise is highly original and thought provoking.
One of my main areas of concern regarding the paper is the complete of lack of consideration of private/corporate sector food system actors in the discussion - in terms of their role, contribution, responsibility, accountability etc. to issues of food insecurity (for example food retailers are key gatekeepers to the modern food system) and what this means in the context of the paper. I see this as an important, but relatively easily addressed, oversight in the paper as it currently stands.
A second important issue, I feel, relates to the discussion on the usefulness of a human rights framework in addressing food insecurity and food poverty. I think the practical impediments and challenges associated with the implementation of a such an approach in high income countries are not sufficiently addressed in the paper and these are issues which warrants further elaboration. At the same time, I think community-led initiatives in other human rights settings (such as health and housing - evidenced by the success of initiatives such as the Participation and Practice of Rights Project in Northern Ireland) - hold many useful lessons and exemplars for activists engaged with the issue of food security/food poverty and could be highlighted in this regard - and there may be many others closer to home for the authors.
Author Response
One of my main areas of concern regarding the paper is the complete of lack of consideration of private/corporate sector food system actors in the discussion - in terms of their role, contribution, responsibility, accountability etc. to issues of food insecurity (for example food retailers are key gatekeepers to the modern food system) and what this means in the context of the paper. I see this as an important, but relatively easily addressed, oversight in the paper as it currently stands.
We thank the reviewer for this important insight and agree that the corporate sector has a role to play with respect to food security issues across the food chain. However, we think that this requires a separate paper to discuss what this role may be and how academia can work with the corporate sector (ethically) as well as how the corporate sector can work with citizens as opposed to “consumers” in order to really take into consideration their lived experience – where profit may not necessarily be the driving influence. This paper is framed from the perspective of academia, policy makers and non-government organisations. To highlight our particular focus, we added a sentence about this in the introductory section. We would be keen to progress this as a separate paper.
A second important issue, I feel, relates to the discussion on the usefulness of a human rights framework in addressing food insecurity and food poverty. I think the practical impediments and challenges associated with the implementation of a such an approach in high income countries are not sufficiently addressed in the paper and these are issues which warrants further elaboration.
Thank you for this recommendation. We have added some of the challenges to the adoption of the rights framework and added citations.
At the same time, I think community-led initiatives in other human rights settings (such as health and housing - evidenced by the success of initiatives such as the Participation and Practice of Rights Project in Northern Ireland) - hold many useful lessons and exemplars for activists engaged with the issue of food security/food poverty and could be highlighted in this regard - and there may be many others closer to home for the authors.
Thank you for this recommendation. We have added the PPR and several other organizations to expand this section to make it more meaningful and grounded.
Reviewer 2 Report
Comments and Suggestions for Authors RE: Commentary Gallegos and Chilton
Overarching feedback
Thank you for the opportunity to review this important and thought provoking contribution. As per below, I think it is important to clarify that these principles aim to partly redress our failure to prevent food insecurity and tackle its causal factors. There are other substantial obstacles that you are no doubt familiar with, that inhibit success. I suggest section 4 would benefit from a few additional sentences in each area to clarify how these principles could be actioned – for eg: reference examples or draw from WtH / GFNA where these were enacted. Best wishes in taking this work forward.
Minor comments and suggestions
1. ‘Principles’ in title (capital P after : )
2. Line 33, p1. ‘The result is limited access…’ for who? I suggest here you clarify here who these limitations impact disproportionally
3. Line 36-9, p1. This is a long sentence. Consider revising.
4. Line 39, p.1. ‘The lack of success in addressing food and nutrition insecurity is partly attributable to a serious and ever-widening gap between supposed knowledge sitting with the “experts” and the realities of people who know the struggle of food insecurity’. Consider revising as I think this is the underpinning tenet of this commentary.
5. Line 153, p.4. ‘For Witnesses however, the insistence of members on the importance…’
6. Line 270, p.7 Consider using Marmot’s full name as you appear to use full names elsewhere in paper. Ensure this is addressed throughout. For eg: Line 415, p.10
7. In the concluding paragraph, whilst strong, I believe you introduce some new material here. Pending the style guide for a Commentary, this typically breaks convention. I suggest lines 437-443 could be well placed above section 6.
8. Line 449, p.12. Acknowledgements instructional text needs to be removed.
Author Response
Thank you for the opportunity to review this important and thought provoking contribution. As per below, I think it is important to clarify that these principles aim to partly redress our failure to prevent food insecurity and tackle its causal factors.
Thanks for this recommendation, we make this failure more explicit throughout.
There are other substantial obstacles that you are no doubt familiar with, that inhibit success. I suggest section 4 would benefit from a few additional sentences in each area to clarify how these principles could be actioned – for eg: reference examples or draw from WtH / GFNA where these were enacted.
--Thank you for this recommendation. We added such examples with references in each segment.
Best wishes in taking this work forward.
Minor comments and suggestions
1. ‘Principles’ in title (capital P after : )
-- Fixed
2. Line 33, p1. ‘The result is limited access…’ for who? I suggest here you clarify here who these limitations impact disproportionally
-- Fixed
3. Line 36-9, p1. This is a long sentence. Consider revising.
-- Fixed
4. Line 39, p.1. ‘The lack of success in addressing food and nutrition insecurity is partly attributable to a serious and ever-widening gap between supposed knowledge sitting with the “experts” and the realities of people who know the struggle of food insecurity’. Consider revising as I think this is the underpinning tenet of this commentary.
-- Fixed
5. Line 153, p.4. ‘For Witnesses however, the insistence of members on the importance…’
-- Fixed
6. Line 270, p.7 Consider using Marmot’s full name as you appear to use full names elsewhere in paper. Ensure this is addressed throughout. For eg: Line 415, p.10
-- Fixed throughout
7. In the concluding paragraph, whilst strong, I believe you introduce some new material here. Pending the style guide for a Commentary, this typically breaks convention. I suggest lines 437-443 could be well placed above section 6.
We are not clear where the new material is located and believe that what we have indicated is a summary of the content above, albeit in slightly different wording.
8. Line 449, p.12. Acknowledgements instructional text needs to be removed.
-Fixed
Reviewer 3 Report
This article raises very interesting and important points, and is written in an engaging style.
However, there are a number of points (detailed below) where more details and background information are needed in order to strengthen the arguments being made. The article is often critical without proposing constructive changes, or even fully explaining the critique.
In addition, the authors seem to assume, and perpetuate, a clear dichotomy between the ‘expert researcher’ and the ‘expert through lived experience’. This negates the experiences of those who are both, and further perpetuates the stereotype of the disenfranchised poor vs the rich and privileged ‘expert’. This is a very important issue, and the lack of researchers in academia from impoverished/less privileged backgrounds is a serious problem, as pointed out in this paper. However, the authors should be careful not to write in such a way that assumes that ALL academic readers will share their privileged background, as this only further entrenches the idea that academia is not open to those from disadvantaged backgrounds. Some slight editing would be sufficient to highlight the fact that such researchers CAN and DO exist, and may bring particularly useful skills and knowledge to their research.
This article requires some revision in order to strengthen the important arguments and points which are raised – with such strengthening, this article has the potential open a very useful conversation among food security researchers.
Notes
Lines 39 – 42: this is a very strong statement, without any referencing to back it up. It would strengthen your argument if you included some details of the particular lack of success you are alluding to, and why this is due to a widening gap between ‘experts’ and lived experience, rather than, say, a lessening of ‘expert’ influence on policy making.
Line 85: ‘and’ – should this be ‘an’?
Line 86: ‘what’ – should this be ‘was’?
Lines 84 – 88: More detail on this programme would be very useful. For example, what were some of the issues identified by both community and health professionals? How was it identified that the underlying premise was that refugees were doing something wrong? Linking your example at the end of the paragraph with the above would make for a much more compelling case.
Line 124: ‘many researchers’ – back this up with references?
Lines 202 – 204: again, this would be strengthened by some concrete examples.
Lines 230 – 231: ‘lacks connection to human, social, political, ecological and cultural context.’ Perhaps you could propose a revised definition, or at least give some examples of the types of issues which ought to be comprehended in a revised definition. As it is, this is vague, and critical without any constructive changes proposed.
Lines 238 – 244: this is a very important point, and it would benefit from more figures to back up what you are saying here. Can you get any statistics, or any estimates, of the numbers of academics from food insecure/impoverished/discriminated against backgrounds? If not, it might be worth pointing out that these don’t exist, and need to be measured so that we can understand the full extent of this issue.
Lines 246 – 249: excellent point and well made
Lines 289 – 292: again, a very strong statement made without any backing evidence. Please provide some citations or examples which show that this is the case, or rephrase. Particularly as academics are not well known for keeping to a 9-5 schedule…
Author Response
The article is often critical without proposing constructive changes, or even fully explaining the critique.
Thanks for this insight. We have added constructive changes and examples of such. We were also quite limited in space/word limit, and recognize that we are taking on a huge topic which we cannot address fully. We have worked to streamline throughout
In addition, the authors seem to assume, and perpetuate, a clear dichotomy between the ‘expert researcher’ and the ‘expert through lived experience’. This negates the experiences of those who are both, and further perpetuates the stereotype of the disenfranchised poor vs the rich and privileged ‘expert’. This is a very important issue, and the lack of researchers in academia from impoverished/less privileged backgrounds is a serious problem, as pointed out in this paper. However, the authors should be careful not to write in such a way that assumes that ALL academic readers will share their privileged background, as this only further entrenches the idea that academia is not open to those from disadvantaged backgrounds. Some slight editing would be sufficient to highlight the fact that such researchers CAN and DO exist, and may bring particularly useful skills and knowledge to their research.
Thank you for pointing out this major weakness in the paper, and identifying where our approach was myopic and classist. Our sincerest apologies for this mistake. We have sought to fix this limitation throughout, and we will also work to ensure we do not make such a mistake in our future work.
This article requires some revision in order to strengthen the important arguments and points which are raised – with such strengthening, this article has the potential open a very useful conversation among food security researchers.
Thanks. The reviewer comments have helped improve this.
Notes
Lines 39 – 42: this is a very strong statement, without any referencing to back it up. It would strengthen your argument if you included some details of the particular lack of success you are alluding to, and why this is due to a widening gap between ‘experts’ and lived experience, rather than, say, a lessening of ‘expert’ influence on policy making.
--Thanks for the recommendation, we have added more context to this section.
Line 85: ‘and’ – should this be ‘an’?
Fixed
Line 86: ‘what’ – should this be ‘was’?
Fixed
Lines 84 – 88: More detail on this programme would be very useful. For example, what were some of the issues identified by both community and health professionals? How was it identified that the underlying premise was that refugees were doing something wrong? Linking your example at the end of the paragraph with the above would make for a much more compelling case.
We have added some additional information to strengthen this paragraph
Line 124: ‘many researchers’ – back this up with references?
We added references of review articles and edited volume on SNAP
Lines 202 – 204: again, this would be strengthened by some concrete examples.
We added a reference, but refrain from adding even more details as we are up against word limit.
Lines 230 – 231: ‘lacks connection to human, social, political, ecological and cultural context.’ Perhaps you could propose a revised definition, or at least give some examples of the types of issues which ought to be comprehended in a revised definition. As it is, this is vague, and critical without any constructive changes proposed.
--Thanks. We added detail.
Lines 238 – 244: this is a very important point, and it would benefit from more figures to back up what you are saying here. Can you get any statistics, or any estimates, of the numbers of academics from food insecure/impoverished/discriminated against backgrounds? If not, it might be worth pointing out that these don’t exist, and need to be measured so that we can understand the full extent of this issue.
--thanks for this important point. We added language in this area.
Lines 246 – 249: excellent point and well made
--thank you
Lines 289 – 292: again, a very strong statement made without any backing evidence. Please provide some citations or examples which show that this is the case, or rephrase. Particularly as academics are not well known for keeping to a 9-5 schedule…
-- We clarified this section and added a reference.